# Sustainable Transformation of Two Algal Species of Different Genera to High-Value Chemicals and Bioproducts

**DOI:** 10.3390/molecules29010156

**Published:** 2023-12-27

**Authors:** Flora V. Tsvetanova, Stanislava S. Boyadzhieva, Jose A. Paixão Coelho, Dragomir S. Yankov, Roumiana P. Stateva

**Affiliations:** 1Institute of Chemical Engineering, Bulgarian Academy of Sciences, 1113 Sofia, Bulgaria; florablue@abv.bg (F.V.T.); maleic@abv.bg (S.S.B.); yanpe@bas.bg (D.S.Y.); 2Instituto Superior de Engenharia de Lisboa, Instituto Politécnico de Lisboa, Rua Conselheiro Emídio Navarro 1, 1959-007 Lisboa, Portugal; 3Centro de Química Estrutural, Instituto Superior Técnico, Universidade de Lisboa, Av. RoviscoPais 1, 1049-001 Lisboa, Portugal

**Keywords:** microalgae, *Scenedesmus obliquus* BGP, *Porphyridium cruentum*, supercritical extraction, fatty acids, total phenolic content, antioxidant activity

## Abstract

This study investigates the potential of two algae species from different genera, namely the recently isolated *Scenedesmus obliquus* BGP and *Porphyridium cruentum*, from the perspective of their integral sustainable transformation to valuable substances. Conventional Soxhlet and environmentally friendly supercritical fluid extraction were applied to recover oils from the species. The extracts were characterized through analytical techniques, such as GC-Fid and LC-MS/MS, which allowed their qualitative and quantitative differentiation. Thus, *P. cruentum* oils contained up to 43% C20:4 and C20:5 fatty acids, while those of *S. obliquus* BGP had only residual amounts. The LC-MS/MS analysis of phenolic compounds in the *S. obliquus* BGP and *P. cruentum* extracts showed higher content of 3-OH-4-methoxybenzoic acid and kaempferol 3-*O*-glycoside in the former and higher amounts of ferulic acid in the latter. Total phenolic content and antioxidant activity of the oils were also determined and compared. The compositional analysis of the oil extracts revealed significant differences and varying potentialities based on their genera and method of extraction. To the best of our knowledge our work is unique in providing such detailed information about the transformation prospects of the two algae species to high-value chemicals and bioproducts.

## 1. Introduction

Microalgae, as a biological resource, have drawn considerable interest in the last years, particularly since they were granted the GRAS (Generally Recognized As Safe) status. Consequently, an avenue for using them as sustainable and appealing green factories for valuable compounds of nutritional and health benefits was widely opened. Moreover, unlike any other crop, they are known to have the lowest carbon, water, and arable footprint. Still, compared with other organisms, microalgae are not that well examined and studied, since just a few among the tens of thousands of microalgae species that exist worldwide have been described [1].

In compliance with the concept of circular economy, the development of effective and efficient biorefineries capable of producing ultra-low-footprint and high-value biochemicals from sustainable sources is essential. A biorefinery integrates various processes to transform biomass to high-added-value energy and non-energy-related compounds. Microalgal biomass is an underutilized resource with enormous commercial potential, as compared to terrestrial plants, due to their abundance.

The beneficial biomolecules accumulated by microalgae are widely applicable in different industrial fields such as nutritional, pharmaceutical, medicinal, cosmetic, etc. Among the most widely exploited in industry microalgae are the red ones, especially from the *Porphyridium* genus. They are known to produce several classes of essential compounds like long-chain polyunsaturated fatty acids (PUFA), exopolysaccharides, pigments, antioxidants, and various microelements. PUFAs contain low cholesterol levels and are important substances, especially for human well-being, as they contribute to cardiovascular, eye, and brain maintenance. For example, *Porphyridium* spp. are reported to synthesize omega 3 eicosapentaenoic (EPA, 20:5), docosahexaenoic (DHA, 22:6), and omega 6 arachidonic (ARA, 20:4) acids, which the human body is incapable of producing [1,2].

The recovery of valuable bioactives from microalgae by applying conventional and non-conventional methods has been a topic of extensive research in the past years. The methods examined include traditional extraction with an organic solvent (solid–liquid, Soxhlet, maceration), ultrasound, and microwave assisted extractions, as well as extractions applying compressed fluids, e.g., supercritical CO_2_ (scCO_2_) neat or with a co-solvent, as well as pressurized liquid extraction, etc. Details are published in a number of review articles, for example, [3,4], as well as research papers [2,5,6].

The main aim of our investigation was to compare two algal species of different genera from the perspective of their sustainable transformation to high-value substances with potential applications in the food, nutra-, pharmaceutical, energy, etc., industries. 

As model targets, representatives of the genus *Scenedesmus* (green algae) and the genus *Porphyridium* (red algae) were chosen. 

The selection of the two genera was motivated and substantiated since:-*Scenedesmus* species are freshwater algae, which exhibit high growth rates, multiply quickly, adapt easily to changing conditions, and hence are suitable for semi-industrial and industrial cultivation. These algae are high in nutrition and synthesize biologically active substances such as carotenoids, chlorophylls, mycosporin-like amino acids, and polyphenols, with antioxidant and antiviral potential.-The algae of the genus *Porphyridium* are representatives of a totally different group of marine deep-sea algae, which are characterized by a pigment composition different from that of green algae. Furthermore, they are rich sources of unique polyunsaturated fatty acids, carotenes, phycobiliproteins, amino acids, and minerals such as Ca, Mg, Zn, and K [7].-These genera are interesting and challenging in their own right and provide examples of the generic issues to be faced when realizing any one-feed/multi-product biorefinery targeting the sustainable transformation of a biomass to valuables and bioactives with applications in various industries.

As representatives of the two genera, the green algae *Scenedesmus obliquus* BGP, a recently isolated Bulgarian strain, and the red algae *Porphyridium cruentum* were chosen.

Application of traditional techniques for recovering valuable compounds from *S. obliquus* BGP has been very limited, resulting in insufficient data. To gain insights into their performance potential for obtaining bioactives, secondary metabolites, and other important compounds sustainably, it is crucial to use advanced environmentally friendly techniques like supercritical fluid extraction, the efficiency and viability of which can be compared to traditional methods such as Soxhlet. Thus, new opportunities for obtaining high-quality extracts rich in valuable compounds from *S. obliquus* BGP will be uncovered.

Since the extraction method specificity (including types of solvents/co-solvents applied, etc.) is among the most important factors, the impact of the techniques employed on the yield and on the composition of selected oils was analyzed. 

Thus, the comparison was based on essential indicators such as: (i) fatty acid composition from FAME GC-FID, (ii) identification and quantification of antioxidants by LC-MS/MS, (iii) total phenolic content (TPC) and antioxidant activity (AA), and (iv) several additional parameters and indices. The new data and information obtained allow us to reveal the two strains’ importance and potential for sustainable utilization in the food, food supplements, pharmaceutical, cosmetics, energy, etc., industries. 

To the best of our knowledge, no such detailed comparison of strains belonging to different genera, neither in terms of depth nor extent, as presented by us, has been published before.

## 2. Results and Discussion 

This section starts with a brief comparison of the biochemical composition of the two species with data from the literature. Then, the efficiencies of the conventional Soxhlet and extractions with supercritical CO_2_ (scCO_2_), both neat and with a co-solvent, are compared based on yield. The influence of operating conditions (solvents, temperature, and pressure) is discussed as well. Subsequently, the chemical composition, total phenolic content (TPC), and antioxidant activity (AA) of certain extracts are presented.


*Biochemical composition*


Our previous results [8] demonstrated that the recently identified *S*. *obliquus* BGP synthesizes considerable amounts of proteins, carbohydrates, and lipids. In particular, with regard to the latter, the strain is one of the top lipid producers among the *Scenedesmus* genus species reported in the literature. Other authors have also analyzed the biochemical composition of *S. obliquus*. For example, in [9], the protein, lipids, and carbohydrates percent reported were 52.00, 9.70, and 7.90%, respectively, while the data of Silva et al. [10] showed that they were 37.15, 10.29, 21.93% and for the three biochemical groups. In another work [11], comparing the composition of *S. obliquus* with *Selenastrum bibraianum* the former was proved to have the highest lipid (42.6 ± 0.01%) and carbohydrate content (27.7 ± 0.02%) when cultured in Bristol media under control conditions.

Regarding *P. cruentum*, in their recent contribution, Ardiles et al. [12] compared the strain biochemical composition, determined by them, with the data of [13,14]. For example, the lipids content has been reported in wide ranges: 14.67 ± 0.24 (%, *w*/*w*) [12], 9–14 (% *w*/*w*) [13], and 5.3 ± 0.3 (% *w*/*w*) [14].

The qualitative deviations between the biochemical composition of *P. cruentum* and *S. obliquus* BGP determined by us and those shown in the literature are considerable in some cases. Still, it should be underlined that the amounts of the chemical species identified depend greatly not only on the cultivation conditions but also the strain. For example, it has been demonstrated that *Scenedesmus* spp. can accumulate lipids in the range of 15–40% of their dry matter [3]. However, when high stress levels are inflicted during cultivation, microalgae can accumulate lipids as high as 70–90% of their dry matter.

### 2.1. Extraction Yield

#### 2.1.1. Soxhlet Extractions

The extraction yields achieved by one- and two-step Soxhlet extraction are given in Table 1. Considering the results of the compositional analysis of *P. cruentum*, which showed a low content of lipids, the low yield of the first step Soxhlet with *n*-hexane was expected (Table 1). On the other end is the yield achieved by the second-step Soxhlet with ethanol performed on the *P. cruentum* residual matrix, which was over 1.7-times higher than the respective one for the *S. obliquus* BGP.

Those results are not surprising since, as discussed in [15], the impact of solvent and extraction methods on the various lipidic classes recovery is considerable.

Considering that freshwater algae such as *S. obliquus* are generally rich in neutral and medium-chain PUFAs, like C16 and C18, while marine algae are rich in long-chain polar PUFAs (e.g., C20 and C22), the higher yield obtained from the second step Soxhlet of *P. cruentum* is due to the use of polar GRAS solvent ethanol, which enhances the recovery of polar PUFAs.

#### 2.1.2. Supercritical Fluid Extraction (SFE)

SFE, as an advanced green technique, is widely applied to recover valuables from microalgae. For example, Tzima et al. [4], in their extensive and in-depth review, analyzed over 100 articles reporting SFE of carotenoids, chlorophylls, tocopherols, lipids, and fatty acids from microalgae, with a special section devoted to the genera *Scenedesmus* and certain *S. obliquus* species.

In another recent paper, the efficiencies of conventional, microwave-assisted, and SFE to recover bioactives from *S. obliquus* were compared [9], and it was shown that the results for the solid/liquid and microwave-assisted extractions were comparable, while in the SFE extraction, the lowest yield of bioactives was achieved. On the other hand, the SFE resulted in increased carotenoid content and enhanced antioxidant activity. Gilbert-Lopez et al. [16] examined the SFE by neat CO_2_ of *S. obliquus*, while Guedes et al. [17] investigated the influence of pressure, temperature, CO_2_ flow rate, and a polar co-solvent on the yields of carotenoids and chlorophylls in the SFE of a wild strain of *S. obliquus*. As reported in [18], the SFE extracts were rich in tricylglycerides but with low carotenoids and chlorophyll content in comparison with the gas-expanded liquid extraction [16].

With regard to *P. cruentum* recovery of polyunsaturated fatty acids (PUFAs), and total carotenoids from that strain by scCO_2_ and subcritical *n*-butane was explored by Feller et al. [19]. Ardiles et al. [12] applied conventional methods like maceration and freeze/thaw, as well as microwave and ultrasound, for the recovery of phycoerythrin (PE) from *P. cruentum* and *P. purpureum.* Gallego et al. [5] advocated a green downstream approach to the valorization of *P. cruentum* biomass, applying pressurized liquids.

The yield of *S. obliquus* BGP achieved in our study applying neat scCO_2_, as a function of temperature and pressure, is plotted vs. the extraction time and is displayed in Figure 1.

In all cases studied, the yield was low and varied in the range of 1.12–1.96%. For both pressures applied, the negative influence of temperature was clearly demonstrated—lowest yields were registered at the highest temperature applied of 60 °C. The highest yield with neat scCO_2_ was obtained at 40 °C and 400 bar.

Our results compare reasonably well with the results of Georgiopoulou et al. [9] who performed SFE of *S*. *obliquus* at *T* = (40, 50, 60) °C and *p* = (110; 250) bar and achieved a yield with neat scCO_2_ in the range of 0.98–4.20%, as well with those of [16], where the temperatures applied are analogous to ours but the pressures were varied in the range *p* = (100; 250; 400) bar. The highest yield obtained was 1.15% at *T* = 40 °C and *p =* 400 bar.

SFE with neat scCO_2_ of *P. cruentum* was not performed since the strain showed a low content of lipids, which was confirmed by the very low yield of the first-step Soxhlet with *n*-hexane achieved (Table 1).

Next, the influence of 10% ethanol as a co-solvent on the extraction yield was studied for both species. Based on the results obtained for *S. obliquus* BGP with neat scCO_2_, the operating parameters chosen were *p* = 400 bar and *T* = (40, 60) °C, respectively.

In the case of *S. obliquus* BGP, the yield was increased considerably, and the influence of temperature exhibited a trend opposite to that displayed in the case of neat scCO_2_. Thus, the highest yield (12.29%) was achieved at 60 °C. However, it was about two-times lower than one-step Soxhlet ethanol extraction. At 40 °C, the yield was lower but commensurable with that at 60 °C (10.33%). Georgiopoulou et al. also used 10% ethanol as a cosolvent at 60 °C and 250 bar and reported a yield of 9.75% [9]. The cumulative experimental kinetic extraction curves plotted vs. the extraction time for *S. obliquus* are displayed in Figure 2.

For *P. cruentum,* the influence of pressure was first studied at 40 °C, with 10% co-solvent ethanol and a scCO_2_ flow rate of 1 L/min. As shown (Figure 3), the highest yield registered was 1.93% at *p* = 400 bar, while the lowest was 1.04% at the highest pressure applied. Our initial expectations that the presence of the co-solvent would increase yields to values, if not commensurable then at least not that lower than the Soxhlet ethanol extraction, were not fulfilled. The reasons behind that assumption were that since the strain cellular walls were not as thick as those of *S. obliquus* BGP, the application of relatively high pressures and a co-solvent enhanced the mass transfer, which resulted in higher yield values. Still, the highest yield attained was almost 20-times lower than the one-step Soxhlet ethanol extraction. Gallego et al. [5] used pressurized ethanol (105 bars) at a wide range of temperatures (from 50 to 150 °C) and 20 min extraction time and reported increasing yields with increasing temperature—from 3.12 to 11.36%.

Though a direct comparison with our results cannot be made, the maximum yield registered by us at 10% ethanol is commensurable with the lowest value at 50 °C reported by Gallego et al. [5].

Then, the influence of temperature at 400 bar was examined. Two temperatures were applied, *T* = 40 and 60 °C, respectively, at a 1 L/min scCO_2_ flow rate. The trend was clear-cut—the higher temperature had a negative effect on the yield, as shown in Figure 4. Also, as demonstrated, the two-times-lower scCO_2_ flow rate at 40 °C and 400 bar rendered a lower yield (1.33% vs. 1.96%) and required a longer time to achieve it.

### 2.2. Extracts Quali- and Quantification

#### 2.2.1. GC-FID

Microalgae produce saturated and unsaturated fatty acids. Saturated fatty acids can be bio-transformed to biofuels, while unsaturated fatty acids are used in food, nutraceuticals, pharmaceuticals, cosmetics, etc. Since non-polar solvents (e.g., n-hexane) are best for the extraction of neutral saponifiable lipids, while polar solvents like ethanol perform better in the recovery of more polar lipids, two-step Soxhlet procedure was applied. In the first step, mainly neutral lipids extracted by n-hexane should be recovered and identified in the oils, while ethanol in the second step should extract the polar lipids from the residual biomass. Still, it should be underlined that in microalgal cells, the two types of lipids are not completely isolated, and in many cases, the application of polar solvents leads to the joint extraction of neutral and polar lipids.

The GC-FID results of the *S. obliquus* BGP and *P. cruentum* oils recovered by the two-step Soxhlet are displayed in Table 2. The results from the Soxhlet ethanol extractions showed a higher relative percentage of saturated (SFAs) and a lower percentage of unsaturated (mono-, di- and polyunsaturated—MUFAs, DUFAs, PUFAs) fatty acids in the red microalgae when compared to the green one (Table 2). Palmitic acid (C16:0) was dominant in the SFA percentage, both for the green and red strains.

For the three *S. obliquus* BGP oils analyzed, the percentages of SFAs, and ∑ MUFA + DUFA + PUFA did not differ substantially, while for *P. cruentum*, the lowest relative percentage of SFA (respectively, the highest ∑ MUFA + DUFA + PUFA) was detected in the oil recovered by the first-step Soxhlet n-hexane. Soxhlet with ethanol and second-step Soxhlet for that strain produced more or less commensurable percentages.

A more comprehensive examination of the results shows that the contribution of the MUFA, DUFA, and PUFA in the sum of unsaturated fatty acids was completely different for both strains. For example, the percentage of the nonpolar oleic acid (C18:1 ω9) was the highest contributor to the MUFAs % of S. obliquus. The highest relative percent was achieved by Soxhlet n-hexane, which was over nine-times higher than that of the corresponding *P. cruentum* oil recovered. Consequently, the MUFAs percentage in the *S. obliquus* oils was almost eight-times higher than that detected in the P. cruentum. On the other end is the PUFAs percentage —for P. cruentum, they were the dominant ones—almost three-times higher than the corresponding ones for S. obliquus.

Another interesting detail is that oils of both strains were relatively rich in linoleic acid (LA, C18:2 ω6) with commensurable relative percents in the range of 13.5–15.2%, with the Soxhlet n-hexane *P. cruentum* oil being the only exception, for which about 21% of LA was detected. The percentages of γ-linolenic acid (GLA C18:3 ω6) in all extracts of both strains were quite low, while for alpha linoleic acid (ALA C18:3 ω3), the trend was different—its relative percentages in all *S. obliquus* BGP extracts examined, though not very high, were still from 19 to 33 times higher than the corresponding ones in *P. cruentum* oils.

It is well known that the genus Porphyridium synthesizes the nutritionally important PUFAs eicosapentaenoic acid (EPA, 20:5 ω-3) and arachidonic acid (AA, 20:4 ω-6), the quantities of which, in certain strains, can comprise more than 40% of the total fatty acids.

In the extracts of the particular *P. cruentum* strain we studied, the highest percentages of AA were registered in the oils recovered by the first and second steps of Soxhlet (with n-hexane and ethanol, respectively). It should be noted that the AA percentage identified in the first-step Soxhlet was the second highest in all oils examined, being lower only than the percentage of the C16:0 obtained by Soxhlet ethanol. The relative percentages of EPA were over two-times lower than those of AA. Neither PUFAs were detected in the *S. obliquus* BGP oils.

As known, LA is an essential fatty acid that is not synthesized in the human body and should be provided by food. Its importance is further increased by the fact that humans can synthesize AA from LA. In this sense, both *S. obliquus* BGP and *P. cruentum* can be used as a sustainable source of LA. In addition, AA is important for human cell functioning since its metabolic breakdown leads to an enhanced production of prostaglandin E2, a hormone-like substance which takes part in a variety of bodily functions [17].

Hence, the ability of *P. cruentum* to synthesize AA and EPA (the first in high amounts) defines the strain as a promising alternative to oils from fish and land-based plant sources, since both acids are used to enrich functional foods.

The *S. obliquus* BGP and *P. cruentum* oils recovered by scCO_2_ + 10% ethanol as a co-solvent performed at the operating conditions for which the maximum yield was achieved were also analyzed by GC-FID, and the results obtained are displayed in Table 3.

The *P. cruentum* oil obtained by scCO_2_ + ethanol again showed higher relative percentage of SFAs when compared to the *S. oblicuus* one, with C16:0 being the major contributor.

In complete analogy to the Soxhlet extracted oils, the percentage of oleic acid was the highest contributor to the MUFAs percentage of *S. obliquus*. As should be expected, its percentage was commensurable only with that detected in the second-step Soxhlet + ethanol but considerably lower than the first-step Soxhlet, and again over nine-times higher than that of the corresponding *P. cruentum* oil recovered.

With regard to the important AA and EPA, the relative percentage of the first was lower than that registered by Soxhlet ethanol, while the second was higher. Again, in the *S. obliquus* BGP oils, those acids were not detected. Yet one additional interesting result was that while the PUFAs percentage of *P. cruentum* oils was within the percentages achieved by Soxhlet, for the *S. obliquus*, an almost twice higher percentage of PUFAs was achieved when compared to the Soxhlet (see Table 2 and Table 3, respectively). Concerning DUFA, their percentages were comparable for both species.

Table 2 and Table 3 also show the PUFA: SFA ratios. It should be noted that the DUFAs C18:2 and C20:2 were taken into consideration when calculating those ratios since linolenic acid is an important essential acid, while eicosadienoic acid, though rare, is known to modulate the metabolism of other PUFAs.

It is hypothesized that all PUFAs in the diet can depress low-density lipoprotein cholesterol (LDL-C) and lower serum cholesterol levels, whereas all SFAs contribute to high levels of the latter. The results of a recent clinical study confirm the positive influence of PUFAs on high-density lipoprotein cholesterol (HDL-C) levels and total cholesterol—the levels of the former were increased, while those of the latter decreased [20].

The World Health Organization (WHO) reported the guidelines for a “balanced diet”, in which the suggested ratio of PUFA: SFA is above 0.4 [21]. WHO also recommends that the proportions of SFA, MUFA, and PUFA in dietary fats should be 1:1.5:1 [22] in order to avoid the risk of developing cardiovascular and other chronic diseases.

The proportions calculated for *S. obliquus* BGP oil recovered by scCO_2_ + ethanol were 21.6:33.0:33.3 (1:1.53:1.54), while for P. cruentum, they were 31.7:5.3:63.0 (1:0.17:1.99), resepctively.

Though none of the above comply with the WHO recommendations still the proportions of *S. obliquus* BGP are better..

Table 2 shows that the PUFA: SFA values of *S. obliquus* BGP oils were generally lower (up to about 2.5 times) than the corresponding ones of the red microalgae. For the latter, the highest PUFA: SFA ratio was registered in the oil recovered by the first step Soxhlet n-hexane—almost twice as that of Soxhlet ethanol. The influence of the extraction techniques on the PUFA: SFA ratios is clearly demonstrated when the data displayed in Table 2 and Table 3 are compared. Thus, the PUFA: SFA of the *S. obliquus* oil (Table 3) was over 1.5-times higher than those of the Soxhlet extractions (1.54 vs. 1.03; 1.02; 1.04). Obviously, the latter enhanced the recovery of SFAs, while the SFE enhanced that of PUFAs. The trend observed for *P. cruentum* was similar when the PUFA: SFA ratio of the SFE oil was compared to that of the Soxhlet ethanol and second-step ethanol (1.99 vs. 1.36; 1.71) but was lower than the Soxhlet n-hexane. Other important indices for the characterization of algal oils were calculated and are presented in Table 4 and Table 5, respectively.

For example, the oxidizability, allylic position equivalent, and bis-allylic position equivalent indices (OX, APE, and BAPE) allow the calculation of the oxidation stability index (OSI). The rate of oxidation of fatty acids of oils/biodiesel depends on the number of double bonds per mole and their relative positions.

The corresponding indices for the oils recoverd from both strains were calculated according to [23,24,25] and are shown in Table 4 and Table 5.
(1)OX=(0.02×C18:1+C18:2+2×C18:3)/100
(2)$$\mathrm{APE}=(2\times \left(C18:1+C18:2+C18:3\right))/100$$
(3)BAPE=(C18:2+2×C18:3)/100
(4)OSI=3.91−0.045×BAPE

A high OSI value of an oil indicates that it is stable and can be used for the production of biodiesel without the addition of any antioxidants to enhance the stability during the OSI period, up to which the oil/biodiesel quality would remain unchanged, and the biodiesel must be entirely utilized for engine operation. Following the ASTM standard, oils can be classified as best, moderate, and poor. The best oils are characterized with an OSI ≥ 3 h since they are considered relatively more stable and do not require the addition of antioxidants for stabilization, i.e., their biodiesel is expected to be entirely utilized before OSI expires and degradation begins.

According to Kumar and Sharma who compared the above indices for different microalgal oils, the best-performing oil was the *S. oblicuus* oil (OSI = 3.87), while the OSI indices of other two non-identified Scenedesmus species varied from 1.93 to 2.13 [26].

As demonstrated in Table 4 and Table 5, *S. obliquus* BGP and *P. cruentum* oils belonged to the group of best oils since their OSI values were in the range of 3.89–3.9, regardless of the technique employed to recover them.

Table 4 and Table 5 also show the indices of unsaturation (UI), which reflect the proportion of FAs with different degrees of unsaturation in the total FA composition of a species. Consequently, unlike the PUFA: SFA ratio, which reflects the impact of highly unsaturated FAs, UI highlights the influence of acids with a low degree of unsaturation like MUFAs and DUFAs. In view of this, UI is usually applied to characterize the composition of macroalgal FAs and used as a reference whether the respective macroalgae may be used as alternative sources of high-quality PUFA instead of fish or fish oil.

UI was calculated according to [25]:UI = 1 × (% monoenoics) + 2 × (% dienoics) + 3 × (% trienoics)       + 4 × (% tetraenoics) +5 × (% pentaenoics) + 6 × (% hexaenoics)(5)

As discussed by Chen and Liu [27], the UI value of seaweeds varies widely—from 45 to 368.68. Colombo et al. [28] used the UI to compare macroalgae caught in the cold waters of Canada with species from the temperate waters of South China. They showed that the UI values of the warm water aglae were in the range from 54 to 151, while for the Canadian algae, the UI varied in the range from 174 to 245, respectively [28]. Our results corroborate those findings as the oils of the deep sea species *P. cruentum* recovered, regardless of the techniques applied, had much higher UI indices when compared to those of *S. obliquus* BGP.

Another important index is the Σ hypercholesterolemic fatty acids/Σ hypercholesterolemic fatty acids ratio (h/H), which is related to cholesterol metabolism. Nutritionally, higher h/H values are considered more beneficial for human health. Chen and Liu [27] mentioned in their recent review that initially the h/H index was introduced by Santos-Silva et al. [29] to assess the effect of fatty acids composition of lamb meat on cholesterol, and they pointed out that compared with the PUFA:SFA, the h/H ratio might better reflect that effect on cardiovascular disease [27].

Later, since there was no C12:0 detected in the lamb meat, in order to characterize the relationship between hypocholesterolemic fatty acid (cis-C18:1 and PUFA) and hypercholesterolemic fatty acid [27], the originally proposed formula was extended as follows:h/H = (cis-C18:1 + ΣPUFA)/(C12:0 + C14:0 + C16:0)(6)

In the present study, we applied the formula advocated by Fernandez et al. [30]:h/H = [(Σ (C18:1 n-9, C18:1 n-7, C18:2 n-6, C18:3 n-6, C18:3 n-3, C20:3 n-6, C20:4 n-6, C20:5 n-3, C22:4 n-6, C22:5 n-3 and C22:6 n-3)/Σ (C14:0 and C16:0)(7)
The results obtained are presented in Table 4 and Table 5, respectively.

The highest h/H was calculated for the *S. obliquus* BGP oil recovered by *n*-hexane in the first step of the two-step Soxhlet. The value was higher but still commensurable with the analogous oil extract of *P. cruentum* (3.24 vs. 3.03). Obviously, the over nine-times-higher quantity of C18:1 (n-9) and over 30-times-higher quantity of C18:3 (n-3) in the *S. obliquus* BGP outperformed the C20:4 present in a substantial amount in the *P. cruentum* but was not detected in the *S. obliquus* BGP at all.

The h/H values of the oil extracts of *S. obliquus* BGP and *P. cruentum* recovered by scCO_2_ + 10% ethanol were 3.22 and 2.21, respectively. The *S. obliquus* BGP h/H index was slightly lower than that of the *n*-hexane Soxhlet extraction, while for the *P. cruentum*, the value was decreased more substantially, owing to the higher content of SFA (C16:0) and the lower level of MUFA (C18:1 (n-9)) and DUFA (C18:2 (n-6)) in the SFE extracts.

It should be noted that, as reported by Chen and Liu [27], one of the highest h/H indices was calculated for *Camelina sativa* oil (11.2–15.0). For red seaweed, h/H was 4.22; for shellfish, it ranged between 1.9 and 4.75, except for *Loxechinus albus*, for which h/H = 0.21, a value lower than those of the other species. That could be a result of the fact that the main food source of *Loxechinus albus* is algae. For fish, the h/H is in the range from 0.87 to about 4.83, etc.

Hence, in order to perform a correct comparison of the h/H values, the method advocated in [27] was applied, and the new values calculated for *S. obliquus* BGP and *P*. *cruentum* oils are displayed in Table 6. In all cases examined, the h/H indices of both microalgal species oils were higher than or in the worst case commensurable with the range of h/H indices calculated for the various fish species. The influence of the extraction technique applied was clearly demonstrated for both species—the highest h/H indices were calculated for oils recovered by scCO_2_ + ethanol, with the *S*. *obliquus* BGP oil being the best performer.

As noted above, a reliable comparison with analogous species was difficult as different authors have used different variants of the h/H formula. Moreover, it is not always clearly stated how the oil was recovered (method, solvents, etc.). Thus, in order to compare our data with the results presented in the seminal paper by Matos et al. [14], the h/H of the *P. cruentum* extracts obtained by the different techniques were yet once again recalculated applying the method used in the reference.

The following values for h/H were obtained: 1.38, 2.88, 1.73, and 2.1, respectively, where the order of techniques followed that of Table 4, with the scCO_2_ + ethanol extraction being the last in the row. Matos et al. [14] used the Soxhlet extraction method with petroleum ether applied after acid digestion with 4.0 N HCl for 6 h, and the h/H index for *P. cruentum* was calculated to be 1.9. Our h/H values were of the same order of magnitude, e.g., the oil recovered by the first step Soxhlet *n*-hexane had the best nutritional quality index equal to 2.88, followed closely by that obtained by scCO_2_ + 10% ethanol at 40 °C and 400 bar with h/H = 2.1.

Finally, Table 4 and Table 5 also display the index of atherogenicity (IA), which characterizes the atherogenic potential of fatty acids. As discussed in [27], IA is a more adequate indicator when compared to the PUFA/SFA ratio, which is too general and unsuitable for assessing the atherogenicity of food products. The lowest IA = 0.25 was calculated for the *S. obliquus* oil recovered by scCO_2_ + 10% ethanol, which was lower than the IA indices of the majority of red and brown seaweeds examined. At the same time, it was either commensurable or slightly higher than those of the majority of the green seaweeds reported in the review.

The only exception being Ulva sp., for which an IA = 0.08 was calculated. With regard to P. cruentum, the IA lowest value = 0.33 was calculated for the oil recovered by Soxhlet n-hexane, while for the oil obtained by scCO_2_ + 10% ethanol, IA = 0.46, which, though lower, is still commensurable with the IA value reported for *P. cruentum* oil in [14].

#### 2.2.2. Analysis and Quantification of Antioxidants

A knowledge of the oils’ composition regarding the presence of phenolics was obtained from the LC-MS/MS analyses of the extracts. The results are shown in Table 7.

The presence of constituents of some important groups of antioxidants was tested, namely hydroxycinnamic, caffeoylquinic, and hydroxybenzoic acids, as well as representatives of four subgroups of flavonoids. The quali- and quantification of the *S. obliquus* BGP and *P. cruentum* extracts showed that the oils of both species were not very rich in antioxidants. Furthermore, the quantities of the bioactives identified varied sometimes by orders of magnitude, which demonstrates not only the influence of the specific genus but also the impact of the recovery methods, operating conditions, and solvents applied. For example, the quantities of hydroxycinnamic and caffeoylquinic acid derivatives in *S. obliquus* BGP and in *P. cruentum* were commensurable, with the exception of ferulic and cinnamic acids, which were more abundant in P. cruentum. With regard to hydroxybenzoic acid derivatives, the picture was different. The highest amount in that group was detected for vanillic acid in the *S. obliquus* oil recovered by second step Soxhlet, followed by 3-OH-4-methoxybenzoic acid. Actually, the quantity of the former was the highest found in all oil tests. The respective amounts of those acids in *P. cruentum* were considerably much lower.

With regard to flavonoids, the best repersented among all subgroups was kaempferol-3-O-glycoside. Its amounts in the oils of both species were of similar magnitude, the only exception being the *S. obliquus* BGP oil recovered by scCO_2_ + 10% ethanol, the quantity of which was the second highest among all antioxidants detected in the oils of both species examined.

#### 2.2.3. Total Phenolic Content (TPC) and Antioxidant Activity (AA)

The TPC and AA of the oils of *S. obliquus* BGP and *P. cruentum* are shown in Table 8. TPC is just a quantitative indicator of the total phenols in the oils and does not provide any information about their particular composition. Still, it showed some trends—for example, examining the results for the Soxhlet extractions of *S. obliquus* BGP, it is clear that the largest amount of polyphenols was extracted in the single-step extraction with ethanol (409.59 μg/mg), followed by scCO_2_ + 10% ethanol, with the amount in the oil recovered by the second-step Soxhlet being the lowest. This trend was expected since, in general, phenolics are better recovered by polar solvents.

The DPPH free radical scavenging activity for the species exhibited a somewhat different pattern; namely, the highest value was measured for the oil recovered by scCO_2_ + ethanol. It was, however, commensurable with that for Soxhlet ethanol. The best (lowest) IC50 value = 1.42 was determined for the oil obtained by the second-step Soxhlet. That can be explained, to a certain extent, by the presence in relatively higher amounts of potent antioxidants like vanilic and 3-OH-4-methoxybenzoic acids.

TPC values calculated for the *P. cruentum* oils were not only lower but the trend of the influence of the extraction methods on the TPC was also different from that for S. obliquus. Thus, the oil obtained by scCO_2_ with 10% ethanol had the highest TPC. Furthermore, in contrast to S. obliquus, the amount of polyphenols in the oil recovered by the second-step Soxhlet ethanol of the spent matrix was higher than the amount registered in the Soxhlet ethanol. On the other hand, the activity toward the DPPH radical for the species was analogous to the one exhibited by S. obliquus. With regard to IC50, the lowest value was calculated for the Soxhlet ethanol extract. The latter was the best among all IC50 values calculated.

Comparing the TPC of the two species, the data clearly show that oils of *S. obliquus* were richer in polyphenols. For both species, the SFE oil extracts had the highest scavenging ability against DPPH free radicals.

In [9], the TPC of *S. obliquus* oil obtained by neat scCO_2_ and scCO_2_ + 10% ethanol were reported. However, a direct comparison cannot be made since the operating conditions are not only different (e.g., lower pressures applied) but also the TPC is represented as mgGA/g_extr_.

The in-depth comprehensive analyses performed on the two species’ oils showed the impact of the techniques’ specificity (operational parameters, nature of solvents, etc.) on their yield and composition. However, the picture of the latter is too complex and reflects not only the effect of the above but also the major influence of the specifics of the strain and the genera, intertwined with other factors like growing conditions, etc., which were not a topic of the present research.

As difficult as it is to generalize our findings, still the following observations and conclusions are valid:i.Both *S. obliquus* BGP and *P. cruentum* oils are “best oils” and can be used for biodiesel production without any antioxidants.ii.*P. cruentum* oils, regardless of the extraction technique used, can enrich functional foods since they have high levels of PUFAs, which are two- to three-times higher than those found in *S. obliquus* BGP oils. *P. cruentum* oil can also serve as a substitute for fish oil because of the high amounts of AA and EPA synthesized.iii.*S. obliquus* BGP and *P. cruentum* oils have commensurable high values of h/H that are in the upper range of the corresponding indices of shellfish and fish. These oils can be used as additives in human nutrition to prevent cardiovascular disease, particularly for people with high blood pressure.iv.*S. obliquus* BGP oil obtained through scCO_2_ extraction with 10% ethanol exhibited the lowest IA of 0.25, which is lower than most red and brown seaweeds as reported in [27]. Therefore, it can be considered a suitable additive to foods or products that can help prevent plaque accumulation and reduce levels of total cholesterol and LDL-C or “bad” cholesterol.v.TPC and AA analysis of two strains’ oils show the impact of genera and extraction methods. *S. obliquus* BGP ethanol oil has the highest TPC, over three-times higher than *P. cruentum*. The lowest IC50 is calculated for *P. cruentum* ethanol oil—over 4.5-times lower than *S. obliquus* BGP. The best AA performers are oils from both species obtained by SFE.

## 3. Materials and Methods

### 3.1. Microalgal Strains

*Scenedesmus obliquus* BGP is a previously not known strain of the genus *Scenedesmus*. It was newly isolated from a rainwater puddle in Sofia, Bulgaria, at an average temperature of 20 °C. The morphological analysis identified the new strain as *Scenedesmus obliquus* (Turpin) Kutzing [8], and the strain was subsequently named *S. obliquus* BGP. A description of the taxonomic and molecular analyses performed was outlined in great detail in a previous contribution by some of the present authors [8] and will not be presented here. Furthermore, it was demonstrated that the strain showed tolerance toward the influence of the most important environmental factors such as light intensity, temperature, and composition of the nutrient medium, and the optimum cultivation conditions were determined.

Lyophilized biomasses of both *S. obliquus* BGP and *P. cruentum* were generously donated to us by the Laboratory of Experimental algology, Institute of Plant Physiology and Genetics, Bulgarian Academy of Sciences.

The monoalgal, non-axenic cultures of red microalga *Porphyridium cruentum* (AG.) (Rhodophyta), strain VISCHER 1935/107, acquired by the Laboratory of Experimental algology, Institute of Plant Physiology and Genetics, Bulgarian Academy of Sciences from the culture collection of the Institute of Botany, Třeboň, The Czech Republic, was grown on the modified culture medium as reported in [31].

The lyophilization of *S. obliquus* BGP and *P. cruentum* biomass was performed in a LGA 05 lyophilizer (Janetzki, Leipzig, Germany), as explained in [8], and then stored in dry–dark conditions prior to use.

### 3.2. Chemicals and Reagents

The chemicals used for the GC–FID analyses were as follows: Supelco 37 Component FAME Mix (CRM47885), reference mixture of fatty acid methyl esters from Sigma-Aldrich (Darmstadt, Germany), toluene (pure, VWR International, Paris, France), sulfuric acid (98%, MerckKGaA, Darmstadt, Germany), sodium chloride (pure, MerkKGaA, Darmstadt, Germany), potassium bicarbonate (pure, VWR International, Paris, France), chloroform (99.8% VWR International, Paris, France), sodium sulfate (pure, Sigma-Aldrich Chemie GmbH, Taufkirchen, Germany), and helium (99.9999%, Air Liquide A/S).

LC MS/MS quantification of polyphenolic compounds used standards enumerated in detail in a previous article [32].

For TPC and antioxidant activity analysis, the following chemicals were used: Folin–Ciocalteu reagent 2 N, sodium carbonate (Merck, Darmstadt, Germany), DPPH (2,2-diphenyl-1-picrylhydrazyl), ATBS, ethanol HPLC grade (Panreac, Barcelona, Spain), gallic acid, Trolox (6-hydroxy-2,5,7,8-tetramethylchroman-2-carboxylic acid) (Sigma-Aldrich, St. Louis, MO, USA), potassium persulfate and absolute ethanol (Neon, Suzano, SP, Brazil), and quercetin dihydrate (Sigma-Aldrich Chemie GmbH, Steinheim, Germany) [32].

The rest of the reagents applied were of the highest purity: methanol ≥ 99.9%, ethanol ≥ 99.8%, and *n*-hexane ≥ 99% were purchased from Honeywell Riedel-de-Haen (Seelze, Germany); ethyl acetate ≥ 99.5% from JLS-Chemie Handel GmbH (Hannover, Germany); methyl tert-butyl ether ≥ 99.8%, and acetonitrile ≥ 99.9% from Sigma-Aldrich (Darmstadt, Germany); and bone dry grade CO_2_ (99.99% pure; no water, Messer, Sofia, Bulgaria).

### 3.3. Biochemical Analyses

In our previous contribution [8], the biochemical composition of *S. obliquus* BGP was analyzed, and the values for proteins (24–45%), lipids (23–30%), and carbohydrates (25–28%) were determined. The biochemical composition of the *P. cruentum* strain VISCHER 1935/107 was reported in [31] to be: proteins (27–38%), lipids (9–12%), and carbohydrates (40–57%). The protocols of the analyses are presented in detail in [8] and will not be reproduced here.

### 3.4. Microalgal Extracts Recovery

#### 3.4.1. Preliminary Preparation of the Material

The lyophilized microalgal samples were firstly crushed additionally to a mean particle diameter (dp) of 0.5 mm and then subjected to ultrasonication in an ultrasonic disintegrator UD 20 (Bandeline electronic, Berlin, Germany) with an ultrasonic field of 35 kHz. After each sonication, the sample treated was examined by a microscope to establish the level of cell disintegration. The procedure was explained in detail in our previous work [8].

#### 3.4.2. Soxhlet Extraction

The protocol for performing the Soxhlet extractions was reported in detail in our previous work [32]. In brief, the Soxhlet apparatus used was ISOLAB NS29/32 + 34/35 (Merck KGaA, Darmstadt, Germany). In the one-step Soxhlet, ethanol was used, while in the two-step, extraction with *n*-hexane was first performed. Subsequently, the residual matrix was subjected to an extraction with ethanol.

In all experiments, the extraction cartridge was filled with 3.0 ± 0.1 g sonicated microalgal biomass (*S. obliquus* BGP or *P. cruentum*). The extraction was performed until a complete discoloration of the solvent was observed. The solvent from the liquid extract was evaporated under vacuum using Hei-VAP Rotary Evaporator (Heidolph Instruments GmbH&Co. KG, Schwabach, Germany). Next, the resulting dry extract was dried at 60 ± 2.0 °C to a constant weight in an air circulation oven, and the yield was calculated according to:(8)$$Yield\left(\%\right)=\frac{mass\;of\;extract\;(g)}{mass\;of\;sample\;(g)}\times 100$$

The extracts were placed in glass vials and kept at 4 °C until analysis. Experiments were performed in triplicates and total extraction yield was expressed as the mean ± standard deviation. The results are presented in Table 1.

#### 3.4.3. Supercritical Fluid Extraction

In our study, *S. obliquus* BGP was extracted with neat scCO_2_ at *T* = (40, 50, and 60) °C and *p* = (400 and 500) bar. The scCO_2_ flow rate was 1 L min^−1^. SCE with a co-solvent ethanol was applied to both strains. The experiments were carried out in a flow apparatus (SFT-110-XW, Supercritical Fluid Technologies Inc., Newark, DE, USA), with two parallel 50 cm^3^ internal volume extractors made from stainless steel tubing (7 cm long, internal diameter 3.02 cm) and temperature controllers for extraction vessels and restrictor valves, which could be adjusted up to 120 °C. The CO_2_ pressure was guaranteed by a SFT Nex10 SCF pump actuate from a compressor model HYAC50-25, Hyundai, South Korea, while in experiments with a co-solvent an additional pump (LL-Class, USA) from SFT, Inc., Newark, DE, USA is used. A full description of the equipment is given in [32] and will not be elaborated further here. In all experiments, about 5 g dry sample of the respective species biomass was placed in the processing vessel.

### 3.5. Characterization and Quantification of the Extracts

The fatty acid composition of certain extracts of *S. obliquus* BGP and *P. cruentum* was determined by GC-FID, the methodology of which was described in detail in [33]. The identification and quantification of the different groups of phenolics by LC-MS/MS was presented exhaustively in its entirety in our earlier works [32].

In addition, the total phenolic content (TPC) of the extracts analyzed was determined using the Folin–Ciocalteu reagent, while the DPPH assay determined the free radical scavenging activity of the samples. The corresponding methodology of the three methods was presented fully in our previously published contribution [32].

## 4. Conclusions

Oils of two algae strains belonging to different genera—the recently isolated Bulgarian strain *S. obliquus* BGP and *P. cruentum*—were extracted using both a conventional method (Soxhlet with hexane and ethanol) and an advanced green technology (scCO_2_ with and without a co-solvent ethanol). The use of scCO_2_ results in lower yields compared to Soxhlet extraction, even with the addition of ethanol as a co-solvent, which is consistent with previous findings in the literature.

The quali- and quantitative analysis of the oils through GC-Fid and LC-MS/MS, as well as the determination of parameters and indices that made it possible to outline their viability, demonstrated the higher potential of *P*. *cruentum* algae as a sustainable source of bioactive compounds with possible applications in the food and/or pharmaceutical industry when compared to the *S*. *obliquus* BGP.

The fatty acid profiles of the oils of the two species differed significantly. In the case of *S. obliquus* BGP, the percentage of different C18 fatty acids (both saturated and unsaturated) ranged from 56.3% to 64.4%, a percentage much higher than that of *P. cruentum*, for which those values were between 19.3% and 28.3%. On the other hand, the latter contained up to 43% of C20:4 and C20:5 fatty acids, which were not detected in the *S. obliquus* BGP. The OSI values calculated for both *S. obliquus* BGP and *P. cruentum* oils position them among the best oils for the production of biodiesel.

Analysis of the fatty acids and polyphenols in *P*. *cruentum* oil indicates its superior potential for food and pharmaceutical applications compared to the green algae. However, the h/H indices calculated for the oils of both species show that they have the capacity to serve as additives to human nutrition. Hence, the algae of both species exhibit promising properties and could be exploited in the future in a one-feed, multi-product biorefinery with a wide variety of applications.

## Figures and Tables

**Figure 1 molecules-29-00156-f001:**
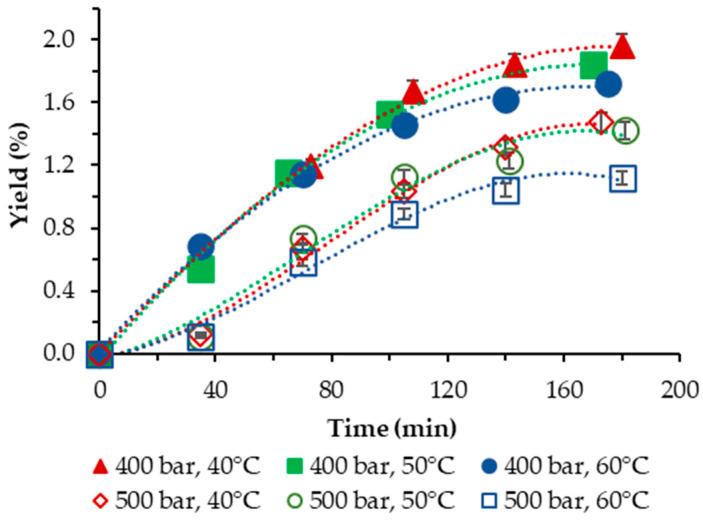
Cumulative extraction yield curves representing the influence of temperature and pressure on the S. *obliquus* BGP yield as a function of time.

**Figure 2 molecules-29-00156-f002:**
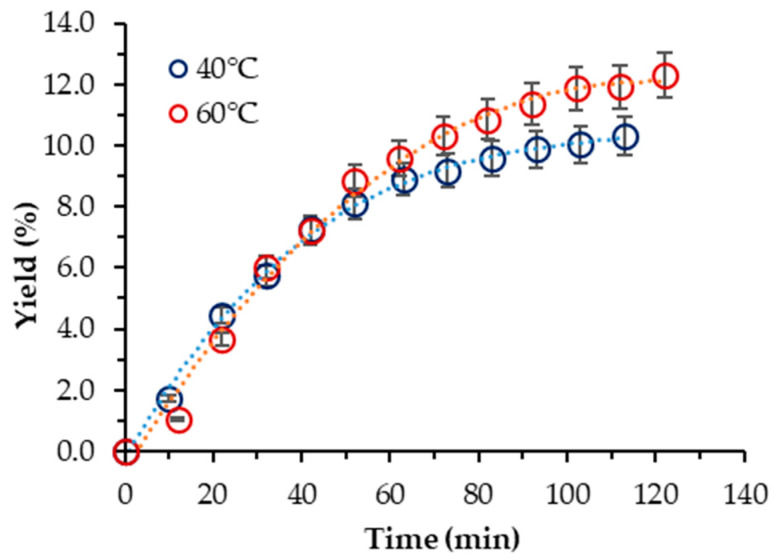
Cumulative extraction yield curves representing the influence of temperature at 400 bar and 10% cosolvent ethanol on the *S. obliquus* BGP yield, as a function of time.

**Figure 3 molecules-29-00156-f003:**
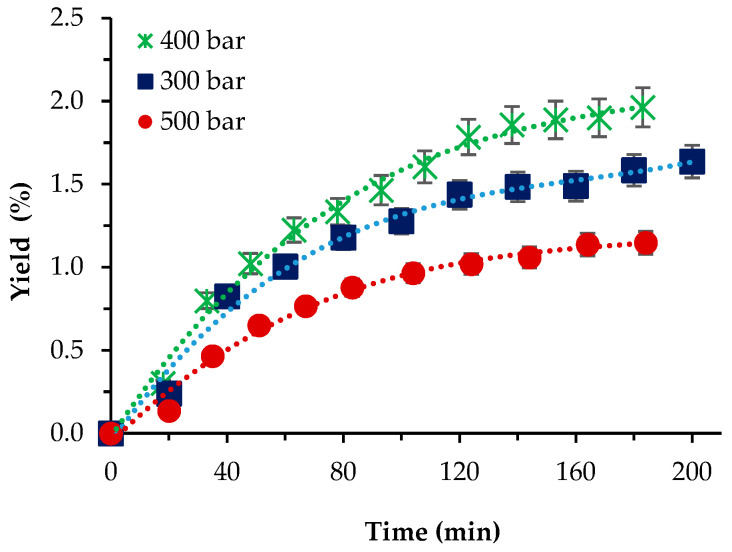
Cumulative extraction yield curves representing the influence of pressure at 40 °C, 10% cosolvent ethanol, and 1 L/min flow rate on the *P. cruentum* yield as a function of time.

**Figure 4 molecules-29-00156-f004:**
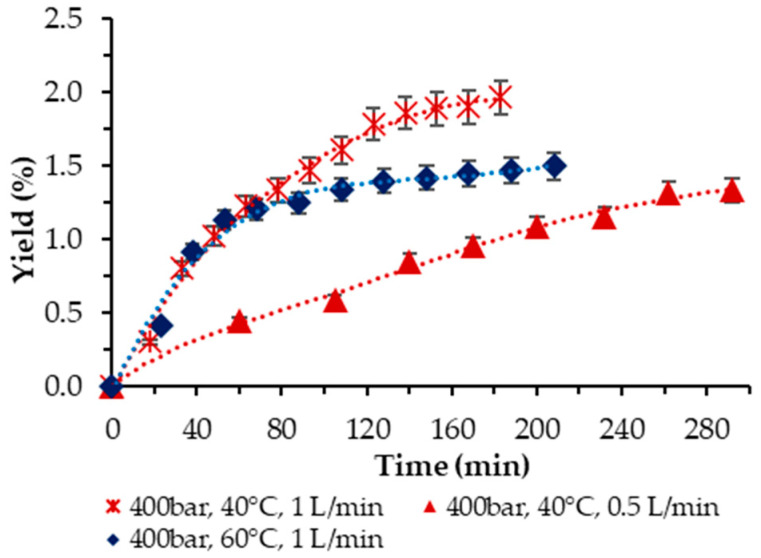
Cumulative extraction yield curves show the effect of temperature at 400 bar and 10% ethanol on *P. cruentum* yield over time, with flow rate variation.

**Table 1 molecules-29-00156-t001:** Influence of the type of Soxhlet method and solvent on the extraction yield of the two algal species.

Extraction Method	Extraction Conditions		*S. obliquus* BGP	*P. cruentum*
	Solvent	Temperature (°C)	Extraction Yield (wt %)	Extraction Yield (wt %)
Soxhlet	ethanol	78	23.6 ± 1.1	26.5 ± 1.15
Two-step Soxhlet	*n*-hexane (step 1) ethanol (step 2)	68 78	11.1 ± 0.5 16.9 ± 0.84 Cumulative yield: 28	2.57 ± 0.14 29.52 ± 1.47 Cumulative yield: 32

Extraction yield expressed in wt % (mean ± standard deviation).

**Table 2 molecules-29-00156-t002:** Fatty acid composition from FAME GC-FID analysis of *S. obliquus* BGP and *P*. *cruentum* Soxhlet oil extracts, expressed as relative percent of total fatty acids identified.

Fatty Acids	Soxhlet extraction					
	*S. obliquus* BGP			*P. cruentum*		
	96% EtOH	First Step *n*-Hexane	Second Step 96% EtOH	96% EtOH	First Step *n*-Hexane	Second Step 96% EtOH
C12:0	traces	1.0 ± 0.07	traces	nd	0.1 ± 0.01	nd
C14:0	0.2 ± 0.01	0.5 ± 0.03	0.2 ± 0.01	0.3 ± 0.02	0.3 ± 0.01	0.2 ± 0.01
C15:0	0.1 ± 0.01	0.1 ± 0.01	0.1 ± 0.01	0.3 ± 0.03	0.2 ± 0.01	0.2 ± 0.02
C16:0	22.3 ± 0.7	18.8 ± 0.6	24.1 ± 0.9	37.8 ± 0.9	23.6 ± 0.7	33.9 ± 0.8
C16:1-isom	5.0 ± 0.5	3.8 ± 0.1	5.7 ± 0.6	1.7 ± 0.1	1.2 ± 0.09	1.6 ± 0.1
C16:2	3.1 ± 0.1	2.6 ± 0.2	3.6 ± 0.5	nd	0.2 ± 0.01	0.2 ± 0.01
C16:3	3.0 ± 0.4	2.4 ± 0.2	3.5 ± 0.4	nd	nd	nd
C16:4	2.4 ± 0.2	1.7 ± 0.1	3.0 ± 0.3	nd	nd	nd
C17:0	0.2 ± 0.02	0.2 ± 0.01	0.3 ± 0.02	0.2 ± 0.01	0.2 ± 0.02	0.1 ± 0.01
C18:0	2.7 ± 0.3	3.2 ± 0.4	2.2 ± 0.3	1.0 ± 0.1	0.9 ± 0.03	0.6 ± 0.02
C18:1 (n-9)	32.8 ± 0.7	38.7 ± 0.9	27.6 ± 0.8	3.5 ± 0.4	4.5 ± 0.5	2.5 ± 0.3
C18:1 (n-7)	0.7 ± 0.02	0.6 ± 0.01	0.8 ± 0.03	1.3 ± 0.5	0.7 ± 0.03	0.7 ± 0.03
C18:2 (n-6)	14.3 ± 0.5	13.5 ± 0.4	14.8 ± 0.4	15.2 ± 0.5	21.2 ± 0.5	14.9 ± 0.6
C18:3 (n-6)	0.8 ± 0.04	0.6 ± 0.02	1.0 ± 0.04	0.2 ± 0.01	0.5 ± 0.03	0.3 ± 0.03
C18:3 (n-3)	9.5 ± 0.8	9.1 ± 0.7	9.9 ± 0.8	0.5 ± 0.02	0.3 ± 0.01	0.3 ± 0.01
C18:4	1.9 ± 0.1	1.9 ± 0.2	2.0 ± 0.2	nd	nd	nd
C20:0	0.1 ± 0.01	0.1 ± 0.01	0.1 ± 0.01	nd	nd	nd
C20:1	0.3 ± 0.01	0.4 ± 0.03	0.4 ± 0.01	0.2 ± 0.01	nd	nd
C20:2	nd	nd	nd	2.6 ± 0.3	1.0 ± 0.4	1.7 ± 0.5
C20:3-isom	nd	nd	nd	1.9 ± 0.3	2.2 ± 0.7	1.5 ± 0.4
C20:4	nd	nd	nd	23.6 ± 0.8	34.8 ± 0.9	31.4 ± 0.9
C20:5	nd	nd	nd	9.7 ± 0.6	8.1 ± 0.5	9.9 ± 0.9
C22:0	0.2 ± 0.01	0.6 ± 0.04	0.3 ± 0.01	nd	nd	nd
C22:1	0.4 ± 0.01	0.2 ± 0.01	0.4 ± 0.02	nd	nd	nd
SFA	25.8	24.5	27.3	39.6	25.3	35.0
MUFA	39.2	43.7	34.9	6.7	6.4	4.8
DUFA	17.4	16.1	18.4	17.8	22.4	16.8
PUFA	17.6	15.7	19.4	35.9	45.9	43.4
PUFA:SFA	1.03	1.02	1.04	1.36	2.69	1.71

nd—not detected.

**Table 3 molecules-29-00156-t003:** Fatty acid composition from FAME GC-FID analysis of *S. obliquus* BGP and *P. cruentum* scCO_2_ + ethanol oil extracts, expressed as the relative percent of total fatty acids identified.

Fatty Acids	*S. obliquus* BGP	*P. cruentum*
	400 Bar, 40 °C, 10% EtOH	400 Bar, 40 °C, 10% EtOH
C12:0	nd	0.7 ± 0.03
C14:0	0.2 ± 0.01	0.4 ± 0.01
C15:0	0.1 ± 0.01	0.3 ± 0.02
C16:0	18.6 ± 0.4	29.0 ± 0.6
C16:1-isom	2.9 ± 0.2	1.2 ± 0.1
C16:2	2.5 ± 0.1	nd
C16:3	3.7 ± 0.3	nd
C16:4	5.9 ± 0.3	nd
C17:0	0.1 ± 0.01	0.2 ± 0.01
C18:0	1.6 ± 0.09	1.1 ± 0.09
C18:1 (n-9)	28.6 ± 0.7	3.3 ± 0.3
C18:1 (n-7)	0.9 ± 0.07	0.8 ± 0.05
C18:2 (n-6)	11.6 ± 0.3	17.5 ± 0.4
C18:3 (n-6)	0.4 ± 0.03	0.3 ± 0.02
C18:3 (n-3)	19.1 ± 0.8	0.2 ± 0.01
C18:4	2.2 ± 0.09	nd
C20:0	0.1 ± 0.01	nd
C20:1	0.3 ± 0.01	nd
C20:2	nd	2.1 ± 0.3
C20:3-isom	nd	2.2 ± 0.3
C20:4	nd	29.0 ± 0.8
C20:5	nd	11.7 ± 0.5
C22:0	0.7 ± 0.03	nd
C22:1	0.3 ± 0.02	nd
C24:0	0.2 ± 0.01	nd
SFA	21.6	31.7
MUFA	33.0	5.3
DUFA	14.1	19.6
PUFA	31.3	43.4
PUFA:SFA	1.54	1.99

nd—not detected.

**Table 4 molecules-29-00156-t004:** Parameters and indices of *S. obliquus* BGP and *P. cruentum* Soxhlet oil extracts.

Oil Parameters and Indices	Soxhlet Extraction					
	*S. obliquus* BGP			*P. cruentum*		
	96% EtOH	First Step *n*-Hexane	Second Step 96% EtOH	96% EtOH	First Step *n*-Hexane	Second Step 96% EtOH
OX	0.26	0.34	0.37	0.17	0.23	0.16
APE	1.16	1.25	1.08	0.41	0.54	0.37
BAPE	0.35	0.33	0.37	0.17	0.23	0.16
OSI	3.89	3.90	3.89	3.90	3.90	3.90
UI	131.1	126.6	134.9	193	239.9	219.8
h/H	2.58	3.24	2.23	1.47	3.03	1.8
IA	0.31	0.29	0.34	0.65	0.33	0.53

**Table 5 molecules-29-00156-t005:** Parameters and indices of S. *obliquus* BGP and *P. cruentum* oils obtained by SCE at 400 bar, 40 °C, and 10% ethanol.

Oil Parameters	*S. obliquus* BGP	*P. cruentum*
	400 Bar, 40 °C, 10% EtOH	400 Bar, 40 °C, 10% EtOH
OX	0.51	0.19
APE	1.21	0.45
BAPE	0.51	0.19
OSI	3.89	3.9
UI	163.2	227.1
h/H	3.22	2.21
IA	0.25	0.46

**Table 6 molecules-29-00156-t006:** h/H indices of *S. obliquus* BGP and *P*. *cruentum* oils calculated.

Species	Extraction Method	h/H = (*cis*-C18:1 + ΣPUFA)/(C12:0 + C14:0+ C16:0)
*S. obliquus* BGP	Soxhlet 96% ethanol	2.27
	Soxhlet *n*-hexane (first-step)	2.71
	Soxhlet 96% ethanol second step after *n*-hexane	1.96
	400 bar, 60 °C, 10% ethanol	3.23
*P. cruentum*	Soxhlet 96% ethanol	1.07
	Soxhlet *n*-hexane (first-step)	2.13
	Soxhlet 96% ethanol second step after *n*-hexane	1.37
	400 bar, 40 °C, 10% ethanol	1.58

**Table 7 molecules-29-00156-t007:** LC–MS/MS analysis of phenolic compounds in *S. obliquus* BGP and *P. cruentum* extracts obtained by Soxhlet ethanol and SFE with co-solvent.

Compound Identified	*S. obliquus* BGP			*P. cruentum*		
	Soxhlet 96% Ethanol	Soxhlet 96% Ethanol Second Step after *n*-Hexane	400 Bar, 60 °C, 10% Ethanol	Soxhlet 96% Ethanol	Soxhlet 96% Ethanol Second Step after *n*-Hexane	400 Bar, 40 °C, 10% Ethanol
ng/mg						
Phenolic acids						
Hydroxycinnamic and caffeoylquinic acid derivatives						
*o*-coumaric acid	4.171	4.143	3.931	3.470	4.242	4.340
*p*-coumaric acid	0.242	0.148	0.040	0.550	0.061	0.262
*m*-coumaric acid	4.439	4.748	4.571	5.216	2.269	4.613
ferulic acid	0.221	0.458	0.412	17.030	2.164	1.246
cinnamic acid	2.934	1.633	3.275	0.559	9.769	2.026
3-*O*-caffeoylquinic (chlorogenic) acid	0.085	0.415	0.393	2.245	0.764	0.546
Hydroxybenzoic acid derivatives						
gallic acid	0.014	0.072	0.046	0.026	0.041	0.012
vanillic acid	2.880	167.174	14.911	n.d.	9.114	7.287
ellagic acid	3.206	0.471	0.839	0.281	0.394	0.530
gentisic acid	0.226	1.778	0.578	2.519	0.091	0.093
protocatechinic acid	0.602	0.453	0.118	2.265	0.111	0.192
*o*-hydroxybenzoic acid	4.964	5.433	4.459	10.913	3.254	5.702
*m*-hydroxybenzoic acid	0.060	1.279	1.786	0.792	0.463	0.841
syringic acid	0.246	0.063	1.130	2.696	1.122	0.210
3-OH-4-methoxybenzoic acid	2.773	94.075	12.126	2.011	8.290	2.744
Flavonoids						
Flavonols						
quercetin	0.603	0.180	0.724	0.084	0.093	0.160
myrecitrin	0.123	0.012	0.126	0.016	0.045	0.016
myrecitin	0.483	0.356	0.435	0.041	0.266	0.799
rutin	2.700	4.597	4.925	4.025	5.867	4.559
resveratrol	0.174	0.202	0.220	0.232	0.190	0.091
kaempferol	0.607	0.101	1.986	0.163	0.053	0.120
kaempferol-3-*O*-glycoside	31.063	13.782	99.507	22.810	0.974	21.391
fisetin	0.482	0.066	0.105	0.204	0.015	0.024
Flavones						
luteolin	0.648	0.103	0.199	0.100	0.051	0.105
apigenin	0.443	0.049	0.910	0.081	0.039	0.067
Flavan-3-ols						
catechin	n.d.	0.066	0.042	0.052	0.398	0.156
epicatechin	0.052	0.118	0.043	0.059	3.590	0.108
Flavanones						
hisperidin	0.123	0.012	0.003	0.003	0.001	0.004
naringenin	0.166	0.006	0.086	0.040	0.002	0.003

Relative standard deviation (RSD) = 2.3%.

**Table 8 molecules-29-00156-t008:** Total phenolic content, antioxidant activity and IC50 of the extracts obtained from *S. obliquus* BGP and *P. cruentum* by different extraction techniques.

Species	Extraction Method	TPC	DPPH	
		Quercetin eq. [μg/mg]	Trolox eq. [mM]	IC50 mg Extract
*S. obliquus* BGP	Soxhlet 96% ethanol	409.59 ± 7.42	1.81	2.65
	Soxhlet 96% ethanol second step after *n*-hexane	155.85 ± 0.40	1.58	1.42
	400 bar, 60 °C, 10% ethanol	255.73 ± 6.33	1.98	3.52
*P. cruentum*	Soxhlet 96% ethanol	134.40 ± 0.80	1.38	0.31
	Soxhlet 96% ethanol second step after *n*-hexane	162.45 ± 3.40	1.09	-
	400 bar, 40 °C, 10% ethanol	182.09 ± 8.08	1.89	2.72

Relative standard deviation (RSD): RSD_DPPH_ = ±3.01%; RSD_IC50_ = ±1.6%.

## Data Availability

Data are contained within the article.

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
