# Peer review of "Sustainable Transformation of Two Algal Species of Different Genera to High-Value Chemicals and Bioproducts"

_molecules, 2023, doi:10.3390/molecules29010156_

Round 1

Reviewer 1 Report

Comments and Suggestions for Authors

This manuscript aims to compare two algal species of different genera with respect to fatty acid composition, phenolic compounds, total phenolic content, and antioxidant activity. In general, this article is not well written and requires substantial corrections in both grammar and wording. Also, the novelty and scientific quality of this manuscript remains inadequate. The following issues should be taken into account to improve the quality of this article:

1.        The supercritical CO2 extraction condition is different among various treatments, which should make it difficult for comparison of results. For instance, in Tables 3 and 5, a condition of 400 bar, 40°C, and 10% EtOH and 400 bar, 40°C, and 10% EtOH were used. In Table 8, a condition of 400 bar, 60°C, 10% EtOH was used. This inconsistency would make it difficult to draw a conclusion.

2.        In Table 2, DUFA was separated from PUFA for calculation of percentage. According to definition of PUFA, it should include DUFA such as linoleic acid (C18:2), an essential fatty acid in maintaining human health. Thus, the calculation of PUFA:SFA can be questionable as DUFA is excluded. In other words, DUFA should be taken into account when calculating the ratio of PUFA:SFA (Lines 327-332).

3.        Lines 332-335: This statement can be controversial as a high content of PUFA can be susceptible to oxidation producing more free radicals to impair human health. In other words, an appropriate ratio of SFA, MUFA, and PUFA should be controlled to maintain human health (DUFA should be included in PUFA).

4.        What is the definition and significance of OX, APE and BAPE?

5.        What are the identification and quantitation methods of phenolic compounds? Did the authors use 28 standards for identification and quantitation? How many replicates were used? Did the authors run statistical analysis? What is the unit of individual phenolic compound? These points should be clarified to make quantitative data convincible.

6.        In Table 3, C18:3 (7-3) accounts for 19.1±0.8% in S. obliquus BGP, which can be questionable as only 9.1-9.9% was shown with the soxhlet extraction method (Table 2).

7.        In the introduction section, there are four paragraphs dealing with the selection of two genera, which should be deleted (Lines 75-95).

8.        In the results and discussion section, the first paragraph should be deleted (Lines119-125). Also, Lines 232 and 235, the number of carbon atoms is inaccurate. Line 237, n-hexane is a non-polar solvent and not low-polar solvent, Lines 543-576, the conclusion is lengthy and should be rewritten. Line 265, is oleic acid nonpolar?

9.        In the material and method section, there are three paragraphs (Lines 663-697) should be moved to the result and discussion section.

10.     A similar study was published in 2021 (reference 7).

All in all, this manuscript should be rejected due to poor novelty and scientific quality.

Comments on the Quality of English Language

Minor editing of English language required

Author Response

We are attaching our answer.

Thank you for your time.

Reviewer 2 Report

Comments and Suggestions for Authors

The manuscript described the investigation of oil compositions extracted by Soxhlet and supercritical fluid techniques from two algae, Scenedesmus obliquus BGP and Porphyridium cruentum VISCHER 1935/107. The highest amounts of the compounds were identified, and total phenolic content and antioxidant activity were determined. Though the idea was not new, and the compositional studies of two algae were studied previously, the results of two different strains could be recognized as records for further applications. The manuscript was well-organized and described. It is recommended after the following suggestion was adopted.

 The main problem is that the study needs more novelty. Since it passed the first evaluation on the desk, I cannot suggest rejection because of the reason that is lacked novelty. I can only remind the editors of the fact. 

Minor:

It is suggested to change the Sample Availability to strain request is available. The procedures can be described because the journal is open-access.

Author Response

(The authors gave the same response as above.)

Reviewer 3 Report

Comments and Suggestions for Authors

The manuscript is well structured, results deep discussed and sustained by a robust methodology.

Authors should handle the following issues:

Line 182, 199, 389 remove the year after the author

Line 212 remove the year after the author and add the corresponding reference in square brackets (as correctly done in line 219)

Line 289, 320 EPA instead of ERA

Line 290, 416 S. obliquus in italics

Line 323, separate PUFAs and %

Line 395, remove italics from the statement

Line 449 after Matos et al add the corresponding reference in square brackets, which is 13

Line 463, after Ulva sp. remove 2

Line 520 amount instead of amonth

Line 598, Obliquus in small

line 678, 680, 683 when you cite for the first time phycoerythrin show its acronym and then report it instead of the full word

Ref 7 year should be in bold

Author Response

We have attached our answers. Thank you for taking the time to review them.

Reviewer 4 Report

Comments and Suggestions for Authors

In this study, authors carried out an environmentally friendly extraction of oils from two algae species from different genera. The extracts were characterized through analytical techniques to quantify the fatty acids and phenolic compounds content. All in all, analysis of the oil extracts revealed qualitative and quantitative differences based on the genera and extraction method. I suggest accepting the manuscript in Molecules journal after minor revision.

1. When studying the influence of 10% ethanol as a co-solvent in the extraction with S. obliquus authors chose 400 bar, and 40 and 60 °C as the operating parameters based on the results obtained with the extraction with neat scCO2. Why did the authors choose 60 ºC if the extraction yield was better with 50 ºC? Additionally, for P. cruentum 40, 50 and 60 ºC were tested.

2. In the case of S. obliquus, the testing with 10% ethanol and scCO2 was performed at a flow rate of 1 l/min (data extracted from Material and Methods), whereas for P. cruentum 0.5 l/min and 1 l/min were tested. There is no apparent reason for this inconsistency. 

3. Some spelling errors have been found in the manuscript. In addition, moderate editing of English language is required.

Comments on the Quality of English Language

Some spelling errors have been found in the manuscript. In addition, moderate editing of English language is required.

Author Response

(The authors gave the same response as above.)

Round 2

Reviewer 1 Report

Comments and Suggestions for Authors

The authors have satisfactorily addressed all the comments raised by reviewers and therefore I recommend acceptance of this article for publication in Molecules.

Comments on the Quality of English Language

Minor editing of English language required